# Seroprevalence of *Toxoplasma gondii* among Sylvatic Rodents in Poland

**DOI:** 10.3390/ani11041048

**Published:** 2021-04-08

**Authors:** Maciej Grzybek, Daniela Antolová, Katarzyna Tołkacz, Mohammed Alsarraf, Jolanta Behnke-Borowczyk, Joanna Nowicka, Jerzy Paleolog, Beata Biernat, Jerzy M. Behnke, Anna Bajer

**Affiliations:** 1Department of Tropical Parasitology, Institute of Maritime and Tropical Medicine, Medical University of Gdansk, Powstania Styczniowego 9B, 81-519 Gdynia, Poland; joanna.nowicka@gumed.edu.pl (J.N.); beata.biernat@gumed.edu.pl (B.B.); 2Institute of Parasitology, Slovak Academy of Sciences, 040 01 Košice, Slovakia; antolova@saske.sk; 3Institute of Biochemistry and Biophysics, Polish Academy of Sciences, 02-106 Warsaw, Poland; k.h.tolkacz@ibb.waw.pl; 4Department of Eco-Epidemiology of Parasitic Diseases, Institute of Developmental Biology and Biomedical Sciences, Faculty of Biology, University of Warsaw, Miecznikowa, 02-096 Warsaw, Poland; muha@biol.uw.edu.pl (M.A.); anabena@biol.uw.edu.pl (A.B.); 5Department of Forest Phytopathology, Faculty of Forestry and Wood Technology, Poznan University of Life Sciences, 60-637 Poznan, Poland; jbehnke@up.poznan.pl; 6Department of Zoology and Animal Ecology, Faculty of Environmental Biology, University of Life Sciences in Lublin, 20-950 Lublin, Poland; jerzy.paleolog@up.lublin.pl; 7School of Life Sciences, University of Nottingham, University Park, Nottingham NG7 2RD, UK; jerzy.behnke@nottingham.ac.uk

**Keywords:** *Toxoplasma gondii*, rodents, seromonitoring, rodent-borne diseases, prevention

## Abstract

**Simple Summary:**

*Toxoplasma gondii* is a significant pathogen affecting humans and animals. Rodents are known to be reservoir hosts of *T. gondii* and therefore play a significant role in the dissemination of this parasite. We conducted seromonitoring for *T. gondii* in four sylvatic rodent species in Poland. We report an overall seroprevalence of 5.5% (3.6% for *Myodes glareolus* and 20% for other vole species). Seroprevalence in bank voles varied significantly between host age and sex. Our results, therefore, make a significant contribution to the understanding of the role of wild rodent populations in the maintenance and dissemination of *T. gondii* and identify key factors that affect the magnitude of seroprevalence in specific host populations.

**Abstract:**

*Toxoplasma gondii* is an intracellular Apicomplexan parasite with a broad range of intermediate hosts, including humans and rodents. Rodents are considered to be reservoirs of infection for their predators, including cats, felids, pigs, and wild boars. We conducted a multi-site, long-term study on *T. gondii* in northeastern Poland. The study aimed to monitor the seroprevalence of *T. gondii* in the four abundant vole species found in the region (*Myodes glareolus, Microtus arvalis, Microtus agrestis*, and *Alexandromys oeconomus*) and to assess the influence of both extrinsic (year of study and study site) and intrinsic (host sex and host age) factors on seroprevalence. A bespoke enzyme-linked immunosorbent assay was used to detect antibodies against *T. gondii*. We examined 577 rodent individuals and detected *T. gondii* antibodies in the sera of all four rodent species with an overall seroprevalence of 5.5% [4.2–7.3] (3.6% [2.6–4.9] for *M. glareolus* and 20% [12–30.9] for *M. arvalis*, *M. agrestis*, and *A. oeconomus*). Seroprevalence in bank voles varied significantly between host age and sex. Seroprevalence increased with host age and was higher in females than males. These results contribute to our understanding of the distribution and abundance of *T. gondii* in voles in Poland and confirm that *T. gondii* also circulates in *M. glareolus* and *M. arvalis*, *M. agrestis* and *A. oeconomus*. Therefore, they may potentially play a role as reservoirs of this parasite in the sylvatic environment.

## 1. Introduction

There is currently considerable interest in understanding not only the transmission of pathogens but also the range of different variables that influence infection dynamics. Among these, both extrinsic factors (e.g., geographic location) and intrinsic factors (e.g., host sex) are known to play varying but crucial roles in exposure of hosts, and their susceptibility, to infection [1,2,3]. Analyzing pathogen dynamics in their wildlife reservoirs is essential in providing a good understanding of their epidemiology and facilitating informed decisions on appropriate measures for their control [4,5,6]. Wild rodents pose a particular threat for human communities because they constitute the most abundant and diversified group of all living mammals [7] and often adopt a synanthropic existence. Although some zoonotic parasites are not transmissible directly from rodents to humans, or pose just a low risk of direct transmission, predation of rodents by carnivore pets may cause infections in these animals, leading to subsequent contamination of households and other human-related environments [8,9].

*Toxoplasma gondii* (*T. gondii*) is an intracellular Apicomplexan parasite with a broad range of intermediate hosts, including humans and rodents [10]. The parasite is present in the tachyzoite stage and changes into bradyzoites, as a result of the conversion of tachyzoites into slow-dividing stages, that eventually form tissue cysts [11]. These can pass from host to host via the food chain and across the placenta resulting in congenitally acquired infections. Rodents are considered to be reservoirs of infection for their predators that include cats, pigs, and dogs. Felid species are the definitive hosts of *T. gondii* and the only hosts that can shed *T. gondii* oocysts into the environment [12].

In 2017, 194 confirmed human congenital toxoplasmosis cases were reported from 22 EU countries. France accounted for 79% of all cases. The number of congenital infections per 100,000 newborns was 5.3 in the European Union and European Economic Area. The highest incidence (number of cases/per 100,000 live births/year) was reported in France (19.9), followed by Slovenia (9.9), Poland (4.5), and Bulgaria (3.1) [13]. In 2018, only 25 cases of congenital toxoplasmosis were reported in Poland, and in 2019 even fewer were reported, with only 14 confirmed cases [14].

We conducted a multi-site, long-term study on *T. gondii* in northeastern Poland. Our objectives were to monitor seroprevalence of *T. gondii* in the four abundant vole species found in the region (*Myodes glareolus*, *Microtus arvalis*, *Microtus agrestis*, and *Alexandromys oeconomus*) and to assess variation in seroprevalence attributable to both intrinsic and extrinsic factors that were quantified.

## 2. Materials and Methods

Our study sites are located in the Mazury Lake District region in north-eastern Poland. The sites and methods used for trapping rodents, and for sampling and processing trapped animals, have all been thoroughly described [15,16,17,18]. Briefly, we used locally constructed Polish wooden live-capture traps. Wheat seeds and peanut butter were used as bate to lure animals into the traps. Trapping was carried out using approximately 300 traps per night, placed in 30–40 m long parallel or divergent lines, 15 m either side of tracks running through the sites and two traps were placed within 2–3 m of one another at each point. Bank voles were sampled in 2002, 2006, and 2010 in three local but disparate forest sites. Other species of voles were sampled only in 2013 from fallow meadows close to one of the forest sites. We examined 577 individuals (*M. glareolus n* = 507; *M. agrestis n* = 10; *M. arvalis n* = 46; and *A. oeconomus n* = 14). Rodent serum was collected and frozen at −72 °C until further analyses.

Serological enzyme-linked immunosorbent assay (ELISA) was used to detect antibodies to *Toxoplasma gondii* in sera. The sensitivity of ELISA method with goat anti-mouse polyvalent antibodies on sera of several rodent species, including *Microtus arvalis* and *Myodes glareolus*, was validated by several studies [19,20,21,22]. Commercially available *T. gondii* antigen (CD Creative Diagnostics, New York, NY, USA) prepared from RH strain [23,24] was used for determination of seropositivity according to Reiterová et al. [21]. Final dilution of antigen was 3.4 μg protein/mL of dilution buffer. Since no positive control sera from *T. gondii* infected *M. arvalis* and *M. glareolus* were available, the cut-off value was determined according to Naguleswaran et al. [25]. The first cut-off value was determined by the mean of all sera on the microtiter plate plus 3 standard deviations (SD). Sera with OD above this value were excluded and the remaining sera were used for calculation of mean absorption of negative samples (M_neg_) and SD_neg_. Sera with OD values above M_neg_ + 4 SD_neg_ were considered to be positive.

The statistical approach has been documented comprehensively in our earlier publications [1,26]. For analysis of prevalence, we used maximum likelihood techniques based on log-linear analysis of contingency tables in the software package IBM SPSS Statistics Version 21 (Armonk, NY, USA). This approach is based on categorical values of the factors of interest, which are used to fit hierarchical log-linear models to multidimensional cross-tabulations using an iterative proportional-fitting algorithm and detect associations between the factors, one of which may be presence/absence of infection. Prevalence values are given in the text and table with 95% confidence limits in square brackets, and in the figure with 95% confidence intervals.

## 3. Results

We examined 577 rodent individuals, and found *T.gondii* antibodies in the sera of all four rodent species, with an overall seroprevalence of 5.5% (Table 1). There was a significant difference in seroprevalence between vole species (*χ*^2^_3_ = 22.58; *p* = 0.001) with *M. arvalis*, *M. agrestis*, and *A. oeconomus* showing 5.6-fold higher seroprevalence (20% [12–30.9]) than *M. glarolus* (3.6% [2.6–4.9]).

In a log-linear model restricted to *M. arvalis*, *M. agrestis*, and *A. oeconomus* there was no difference between these three species (*χ*^2^_2_ = 0.8, *p* = 0.7), nor between the sexes (*χ*^2^_1_ = 0.37; *p* = 0.55) or age classes (*χ*^2^_2_ = 4.67; *p* = 0.097). However, the latter was close to significance and reflected peak seroprevalence in young adult voles (35.7 [15.28–62.89]) in comparison to zero seroprevalence among the youngest voles and 18.0% [9.46–30.87] among the oldest (Figure 1). We therefore confined further analyses to bank voles *(M. glareolus*).

In a log-linear model confined to bank voles, and with year of sampling (3 years) and site (3 sites) taken into account, *T.gondii* seroprevalence differed significantly between the sexes of *M. glareolus* (*χ*^2^_1_ =4.34; *p* = 0.037) and was 3.5 times higher for female (5.5% [3.6–8.1]) compared with male (1.6% [0.7–3.4]) bank voles (Figure 1).

*T. gondii* seroprevalence increased significantly with host age (*χ*^2^_2_ = 11.57; *p* = 0.003), peaking in the oldest bank voles (6.1% [2.77–12.47]), but was lower in bank voles from age class 2 (mostly young adults; 3.5% [1.22–9.50]). No immature bank voles were found to be seropositive. The data in Figure 1 also indicate that seroprevalence increased with host age faster among female bank voles compared with males, although the interaction between age and sex was not significant (*χ*
^2^_2_ = 0.28; *p* = 0.87).

## 4. Discussion

Our results confirm that in the wild *M. glareolus*, *M. arvalis*, *M. agrestis*, and *A. oeconomus* have contact with *T. gondii* and become infected. Therefore, they may potentially play a role as reservoirs of this parasite in the sylvatic environment [27].

According to the meta-analysis carried out by Galeh and colleagues [28], the overall seroprevalence of anti-*Toxoplasma* IgG antibodies in rodents in North America, Australia, and Asia was measured at 5%, 4%, and 4%, respectively. The authors report 1% *T. gondii* seroprevalence in Europe.

Here, we report an overall seroprevalence of *T. gondii* in *M. glareolus*, *M. arvalis*, *M. agrestis*, and *A. oeconomus* of 5.5%, with *M. arvalis*, *M. agrestis*, and *A. oeconomus* showing a considerably higher seroprevalence at 20%. Although the reason for this discrepancy between the rodent genera is not clear, we have noted that cats from local farms are often observed in the meadows where we trapped *Microtus/ Alexandromys* spp., but seldom enter the forests to which bank voles are confined. This is in line with the results obtained with radio-collared farm cats in Switzerland, which kept to a short distance from houses and preferred meadows over forests [29].

Rodents living in an environment with limited populations of cats are less exposed to oocysts [30,31]. Seroprevalence of *T. gondii* in rodents differs significantly between rural and urban environments [32,33]. Urban rodents show higher *T. gondii* seroprevalence than individuals living in the rural environment [22,34,35], reflecting a difference betweeen these environments in the degree of contamination by oocysts from felids [31,36,37].

We found that female bank voles were 3.5 times more likely to be infected with *T. gondii* than males, and, in this respect, our data contrast with reports in the literature in which *T. gondii* seropositivity in male rodents has been recorded generally to be higher than in females [17,28]. It was not unexpected to find in our study that older animals had a higher serological positive rate than juveniles, since the current work was based on the presence/absence of specific antibody against *T. gondii*, reflecting the history of previous infections and not necessarily a current infection [26,38]. Older animals will have had a longer period over which to encounter the infective stages of *T. gondii*, and hence would have experienced greater likelihood of infection than younger individuals [2,9,26].

Transmission of the parasite from wildlife to human habitats may occur especially in rural areas where cats escape human settlements and forage on wild rodents [39,40]. 

## 5. Conclusions

The results presented in this paper provide a significant and novel contribution to our understanding of the seroprevalence rate of *T. gondii* within vole populations. The World Organization for Animal Health recommends assessment of infections in wild rodents to enable effective control and thereby reduction of exposure of domestic animals and humans to zoonotic parasites. However, all appropriate action should be carried out with due regard for animal welfare and biodiversity. Further studies are necessary, therefore, to reveal comprehensively the status of toxoplasmosis in wildlife and to assess the risk of infection for local inhabitants, as well as for visitors to the region.

## Figures and Tables

**Figure 1 animals-11-01048-f001:**
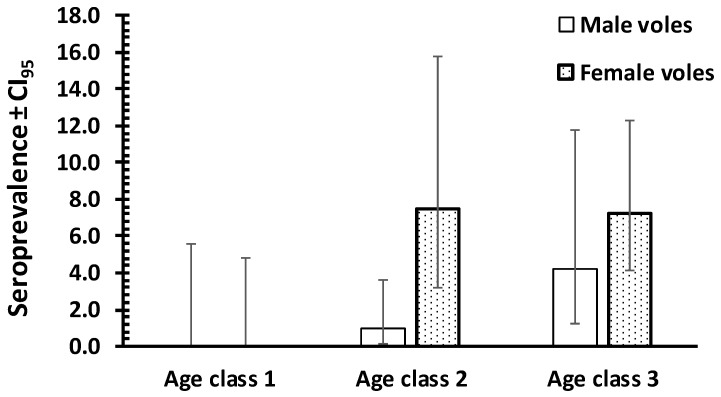
Seroprevalence of *T. gondii* in bank voles in Poland by host sex and by host age class (class 1—immature juvenile voles, *n* = 138; class 2—young adult voles, *n* = 173; and class 3—breeding older animals, *n* = 196). Error bars indicate 95% CI.

**Table 1 animals-11-01048-t001:** Seroprevalence of *T. gondii* in four rodent species in Poland. Seroprevalence is given in percentages with 95% CL in brackets.

Species	Negative	Positive	Total	Seroprevalence (%) ± CL_95_
*M. agrestis*	7	3	10	30.0 [8.7–61.9]
*M. arvalis*	37	9	46	19.6 [9.1–36.8]
*A. oeconomus*	12	2	14	14.3 [2.6–42.6]
*Myodes graelous*	489	18	507	3.6 [2.6–4.9]
Overall	545	32	577	**5.5 [4.2–7.3]**

## Data Availability

Derived data supporting the findings of this study are available from the corresponding author (M.G.) on request.

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
