# Peer review of "Seroprevalence of Toxoplasma gondii among Sylvatic Rodents in Poland"

_animals, 2021, doi:10.3390/ani11041048_

Round 1

Reviewer 1 Report

Toxoplasmosis is caused by the opportunistic protozoan Toxoplasma gondii. It is considered an important food-borne parasitic zoonosis, with a worldwide disease in humans and most warm-blooded animals, including rodents. The present manuscript (MS) describes the seroprevalence of T. gondii Among Sylvatic Rodents in Poland. The manuscript is well written, they carried out a multi-site, long-term study on T. gondii in northeastern Poland. Therefore, I recommend its acceptance for publication after minor reversion.

  1. Line 63: The prevalence lack of unitand pay attention to the unified decimal point throughout the
  2. Line 74: Although these have been described in previous studies, it is best to brief some of the key information of the samples to make it easier to understand.
  3. Line 75: Were serological tests performed in a timely manner after sampling? The earliest samples were taken nearly 20 years. If this is a recent test, authors should take into account false negatives caused by degradation of antibody proteins in the serum.
  4. Figure 1 is not clear, can you change the Figure intoa Table? Please added the number of samples of glareolus in different ages groups.
  5. Line 109: Inyour description, it is difficult for me to determine the positive rate of glareolus of different genders.
  6. Line 132: It should be expressed as “Older animals have higher serological positive rate” instead of “older animals were more likely to have experienced infection than juveniles”, numerous studies have confirmed that young rodents exposed to Toxoplasma are more susceptible to infection.
  7. There are some minor grammatical mistakes and typographical errors in the MS, please cheak it carefuly. For example in line 28, T.gondii should be italic.

Author Response

Reviewer 1

Toxoplasmosis is caused by the opportunistic protozoan Toxoplasma gondii. It is considered an important food-borne parasitic zoonosis, with a worldwide disease in humans and most warm-blooded animals, including rodents. The present manuscript (MS) describes the seroprevalence of T. gondii Among Sylvatic Rodents in Poland. The manuscript is well written, they carried out a multi-site, long-term study on T. gondii in northeastern Poland. Therefore, I recommend its acceptance for publication after minor reversion.

*RESPONSE: We thank Reviewer 1 for her/his comments and expertise.

Line 63: The prevalence lack of unit and pay attention to the unified decimal point throughout the

*RESPONSE: We screened the ms and unified decimals throughout.

Line 74: Although these have been described in previous studies, it is best to brief some of the key information of the samples to make it easier to understand.

*RESPONSE: We developed this section to describe rodent trapping.

Line 75: Were serological tests performed in a timely manner after sampling? The earliest samples were taken nearly 20 years. If this is a recent test, authors should take into account false negatives caused by degradation of antibody proteins in the serum.

*RESPONSE: The serum samples were collected straight after the rodent sampling, frozen at -72’C for further analyses. We put a lot of attention to sample storage and handling. We believe that our work has not been significantly affected by antibody degradation due to time. We avoided samples defrost to minimise antibody degradation. We amended the ms to give clear information that samples were frozen at the capture time and analysed later on.

Figure 1 is not clear, can you change the Figure into a Table? Please added the number of samples of glareolus in different ages groups.

*RESPONSE: We believe that the presented graph clearly shows the host age effect on T. gondii seroprevalence in M. glareolus. We would like to keep the graph instead of changing it into the table. We added the number of animals for each age group in the figure caption.

Line 109: In your description, it is difficult for me to determine the positive rate of glareolus of different genders.

*RESPONSE: We added prevalence value for males and females and amended this section to read:

In a log-linear model confined to bank voles, and with year of sampling (3 years) and site (3 sites) taken into account, T.gondii seroprevalence differed significantly between the sexes of M. glareolus (χ21 =4.34; P=0.037) and was 3.5 times higher for female (5.5% [3.6-8.1]) compared with male (1.6% [0.7-3.4]) bank voles (Figure 1).

Line 132: It should be expressed as “Older animals have higher serological positive rate” instead of “older animals were more likely to have experienced infection than juveniles”, numerous studies have confirmed that young rodents exposed to Toxoplasma are more susceptible to infection.

*RESPONSE: We amended the ms as suggested.

There are some minor grammatical mistakes and typographical errors in the MS, please check it carefully. For example in line 28, T.gondii should be italic.

*RESPONSE: We have screened the ms to search for possible errors. We did our best to write the paper with immaculate English prose.

Reviewer 2 Report

The manuscript Seroprevalence of Toxoplasma gondii Among Sylvatic Rodents in Poland from Grzybek and others is an interesting piece with a good sample size with a poor-studied sample. It can be published in the journal after some revision.

The major issue is the lack of detailed information in the methods section. The authors only cite previous work, which is acceptable, but some basic information must be present in the methods.

In addition, I have other specific comments that need to be addressed.

There is a line dividing the author's affiliation, please remove it.

The symbol in the 2 last authors is not explained. Also, change this symbol.. you may use: † or ⸸ or § (just a style suggestion).

Abstract

extrinsic and intrinsic

Not clear what They mean here.

T. gondii

Not in italic. Please revise it in the entire manuscript.

L33: 20% for other vole species

The seroprevalence was exactly the same for the other species? Not clear. Inform each seroprevalence. 

The abstract is missing the information about how many samples were obtained, this is important.

L33-34: Seroprevalence in bank voles varied significantly between host age and sex.

This sentence has limited meaning. Perhaps you should comment about what you found. Higher age = higher prevalence?  What about differences between species?

L36: Microtus/Alexandromys spp

Why inform the genera here if you mention the species previously?

Introduction

L55: also called tissue cysts

It´s not like that. The tissue cyst contains several bradyzoites.

L62: EU/EEA

Needs a definition. And this sentence needs a citation.

L62: The highest incidence was reported in 62 France (19.9), followed by Slovenia (9.98), Poland (4.48) and Bulgaria (3.13)

The number must be explained (number of cases/100.000 people/year?)

Needs a Oxford comma in the sentence. 

Methods

Although the sampling was already described is not acceptable a methods section so brief without any information at all.

After the sentence at line 74 (have all been thoroughly described [15–18]) the authors must say: Briefly… and describe minimally the sample size for each species, if they use serum or plasma, condensed but basic information must be presented.

Also, please at the beginning of the methods section use the common names followed by the scientific names of all sampled voles, so when you refer to some animals later on, the reader will know about it.

No information about cut-off used (according to the absorbance of each plate control? Please add details about this and add a supplementary table with your raw ELISA data. This is important for an in-house test. Add information about the test validation.

The statistical cannot be described by referring to other papers.    

L80: Prevalence values are given with 95% confidence limits in square 80 brackets

I did not see this in the abstract.

Needs to inform the approval number of the ethics consent. Also, there is a correct place to put such information in the manuscript according to the journal rules.

Results

L90: (χ 2 3

There is a typo here.

L90-91. If you are referring to the species before, keep referring to species, because it may be confusing to the reader.

The sample size per vole species must be informed in the methods section. Also, all the risk factors evaluated must be presented in the methods section.

It's not clear in the results if you compare using the genera or using the species, because there are two species that belong to the same genera, so you merge those?

L96: In a log-linear model restricted to Microtus and Alexandromys spp. there was no difference between these three species

Why restricted to such species? And again, did you merge data from the 2 Microtus species?

L99: age class 2

How the reader will know which age this means? It's informed only later in the figure 1 legend.

Without the information about the statistical methods, I cannot judge it was appropriate. I am sorry, but it´s not acceptable to oblige the reviewer to check other papers online to see what statistical approach was used.

Discussion

L116: M. glareolus and Microtus/Alexandromys spp

Please, keep consistent. Discuses and present results separated into the 4 tested species.

L119: Galeh and colleagues

Need the citation number.

References 1-5 are about other diseases and only reference 10 is about Toxoplasma.

There is too much self-citation (some unrelated) in the paper considering the number of total references used. The authors can add more references about Toxoplasma seroprevalence in wild animals to reduce the % of paper from their group or remove some of the citations. There is much literature about toxoplasma surveys in rodents and wild animals. We run several serological and molecular surveys at the Amazon region for instance.

Author Response

Reviewer 2

The manuscript Seroprevalence of Toxoplasma gondii Among Sylvatic Rodents in Poland from Grzybek and others is an interesting piece with a good sample size with a poor-studied sample. It can be published in the journal after some revision.

The major issue is the lack of detailed information in the methods section. The authors only cite previous work, which is acceptable, but some basic information must be present in the methods.

 *RESPONSE: We thank Reviewer 1 for her/his comments and expertise. The issue concerning method description was also raised by Reviewer 1, and we amended the ms to give a proper methods description.

In addition, I have other specific comments that need to be addressed.

There is a line dividing the author's affiliation, please remove it.

 *RESPONSE: amended as suggested.

The symbol in the 2 last authors is not explained. Also, change this symbol.. you may use: † or ⸸ or § (just a style suggestion).

  *RESPONSE: amended as suggested.

Abstract

extrinsic and intrinsic

Not clear what They mean here.

 *RESPONSE: We provided some explanation to read:

The study aimed to monitor the seroprevalence of T. gondii in the four abundant vole species found in the region (Myodes glareolus, Microtus arvalis, Microtus agrestis and Alexandromys oeconomus) and to assess the influence of both extrinsic (year of study and study site) and intrinsic (host sex and host age) factors on seroprevalence.

T.gondii Not in italic. Please revise it in the entire manuscript.

*RESPONSE: we checked the whole ms to avoid such mistakes

L33: 20% for other vole species

The seroprevalence was exactly the same for the other species? Not clear. Inform each seroprevalence. 

 *RESPONSE: We amended this section to read:

We detected T. gondii antibodies in the sera of all four rodent species with an overall seroprevalence of 5.5% (3.6% for M. glareolus and 20% for M. arvalis, M. agrestis and A. oeconomus).

The abstract is missing the information about how many samples were obtained, this is important.

 *RESPONSE: We provided the number of examined animals.

L33-34: Seroprevalence in bank voles varied significantly between host age and sex.

This sentence has limited meaning. Perhaps you should comment about what you found. Higher age = higher prevalence?  What about differences between species?

  *RESPONSE: We amended the abstract. We clearly showed that seroprevalence increased with host age and was higher in females than in males.

L36: Microtus/Alexandromys spp

Why inform the genera here if you mention the species previously?

   *RESPONSE:  we amended this section to read:

These results contribute to our understanding of the distribution and abundance of T. gondii in voles in Poland and confirm that T. gondii also circulates in M. glareolus and M. arvalis, M. agrestis and A. oeconomus. Therefore, they may potentially play a role as reservoirs of this parasite in the sylvatic environment.

Introduction

L55: also called tissue cysts

It´s not like that. The tissue cyst contains several bradyzoites.

   *RESPONSE: 

We amended the ms to make a clear description of developmental stages to read:

The parasite is present in the tachyzoite stage and changes into bradyzoites, as a result of the conversion of tachyzoites into a slow-dividing stage and form tissue cysts [11].

L62: EU/EEA

Needs a definition. And this sentence needs a citation.

   *RESPONSE:  We provided the whole names from EU and EEA. The citation (ref number 13) is placed after the whole sentence. We amended this section to read:

The number of congenital infections per 100 000 newborns was 5.3 in the European Union and European Economic Area. The highest incidence was reported in France (19.9), followed by Slovenia (9.9), Poland (4.5) and Bulgaria (3.1) [13].

L62: The highest incidence was reported in 62 France (19.9), followed by Slovenia (9.98), Poland (4.48) and Bulgaria (3.13)

The number must be explained (number of cases/100.000 people/year?)

Needs a Oxford comma in the sentence. 

 *RESPONSE: We provide an explanation for numbers in the brackets to read:

The highest incidence (number of cases/ per 100.000 live births/year) was reported in France (19.9), followed by Slovenia (9.9), Poland (4.5) and Bulgaria (3.1) [13]

Methods

Although the sampling was already described is not acceptable a methods section so brief without any information at all.

After the sentence at line 74 (have all been thoroughly described [15–18]) the authors must say: Briefly… and describe minimally the sample size for each species, if they use serum or plasma, condensed but basic information must be presented.

Also, please at the beginning of the methods section use the common names followed by the scientific names of all sampled voles, so when you refer to some animals later on, the reader will know about it.

*RESPONSE: We developed this section to provide a description of rodent trapping.

No information about cut-off used (according to the absorbance of each plate control? Please add details about this and add a supplementary table with your raw ELISA data. This is important for an in-house test. Add information about the test validation.

*RESPONSE: We significantly developed and amended M&M section.

The statistical cannot be described by referring to other papers.    

L80: Prevalence values are given with 95% confidence limits in square 80 brackets

I did not see this in the abstract.

*RESPONSE: We developed this section to provide the description of the statistical approach. We also reported prevalence values with 95% confidence limits within the Abstract.

Needs to inform the approval number of the ethics consent. Also, there is a correct place to put such information in the manuscript according to the journal rules.

*RESPONSE: We moved this section as suggested.

Results

L90: (χ 2 3

There is a typo here.

*RESPONSE: amended as suggested

L90-91. If you are referring to the species before, keep referring to species, because it may be confusing to the reader.

*RESPONSE: amended as suggested

The sample size per vole species must be informed in the methods section. Also, all the risk factors evaluated must be presented in the methods section.

 *RESPONSE: We provided a number of examined individuals per species in the M&M section.

It's not clear in the results if you compare using the genera or using the species, because there are two species that belong to the same genera, so you merge those?

  *RESPONSE: We changed genera into species names along with the whole manuscript. Now it is clear and easy to understand.

L96: In a log-linear model restricted to Microtus and Alexandromys spp. there was no difference between these three species

Why restricted to such species? And again, did you merge data from the 2 Microtus species?

*RESPONSE: This was clarified as described above. We used species name only to make everything clear.

L99: age class 2

How the reader will know which age this means? It's informed only later in the figure 1 legend.

 *RESPONSE: we amended the ms to make sure that the age of rodents is clear for the reader

Without the information about the statistical methods, I cannot judge it was appropriate. I am sorry, but it´s not acceptable to oblige the reviewer to check other papers online to see what statistical approach was used.

*RESPONSE: We developed this section to provide a description of the statistical approach

Discussion

L116: M. glareolus and Microtus/Alexandromys spp

Please, keep consistent. Discuses and present results separated into the 4 tested species.

*RESPONSE: We exchanged “Microtus/Alexandromys spp” into “M. arvalis, M. agrestis and A. oeconomus”

L119: Galeh and colleagues

Need the citation number.

*RESPONSE: we provided the REF number.

References 1-5 are about other diseases and only reference 10 is about Toxoplasma.

There is too much self-citation (some unrelated) in the paper considering the number of total references used. The authors can add more references about Toxoplasma seroprevalence in wild animals to reduce the % of paper from their group or remove some of the citations. There is much literature about toxoplasma surveys in rodents and wild animals. We run several serological and molecular surveys at the Amazon region for instance.

 *RESPONSE: We amended the discussion to provide more references and discuss our results deeper.